# Serum Antibodies Against the E5 Oncoprotein from Human Papillomavirus Type 16 Are Inversely Associated with the Infection and the Degree of Cervical Lesions

**DOI:** 10.3390/biomedicines12122699

**Published:** 2024-11-26

**Authors:** Azucena Salazar-Piña, Minerva Maldonado-Gama, Ana M. Gonzalez-Jaimes, Aurelio Cruz-Valdez, Eduardo Ortiz-Panozo, Fernando Esquivel-Guadarrama, Lourdes Gutierrez-Xicotencatl

**Affiliations:** 1Facultad de Nutrición, Universidad Autónoma del Estado de Morelos, Cuernavaca 62350, Mexico; azucena.salazar@uaem.mx; 2Centro de Investigación Sobre Enfermedades Infecciosas, Instituto Nacional de Salud Pública, Cuernavaca 62100, Mexico; mmaldona@insp.mx; 3Centro de Investigación en Dinámica Celular, Universidad Autónoma del Estado de Morelos, Cuernavaca 62210, Mexico; anamaria.gonzalez@uaem.mx; 4Centro de Investigación en Salud Poblacional, Instituto Nacional de Salud Pública, Cuernavaca 62100, Mexico; acruz@insp.mx (A.C.-V.); eduardo.ortiz@insp.mx (E.O.-P.); 5Facultad de Medicina, Universidad Autónoma del Estado de Morelos, Cuernavaca 62350, Mexico

**Keywords:** anti-E5 antibodies, cervical cancer, HPV16, cervical intraepithelial neoplasia, serology

## Abstract

Background: The humoral immune response against human papillomavirus (HPV) has been suggested as a source of biomarkers for the early detection of cervical cancer (CC). Therefore, we aimed to characterize the antibody response against HPV16 E5 in the natural history of cervical cancer and to determine its usefulness as a biomarker of HPV-associated cervical lesions. Methods: This study was conducted at the Cuautla General Hospital, Morelos, Mexico, with women (18 to 64 years) who agreed to participate. Samples were obtained from 335 women with cervical lesions and 150 women with negative Papanicolaou tests. HPV genotyping was performed by PCR and pyrosequencing, and anti-E5 antibodies were detected by slot blot. Results: The overall anti-E5 antibodies prevalence in the study was 17.9%, with the higher prevalence observed in the no lesion (NL, 49.4%) group, and with a downward trend according to the degree of the cervical lesion, from cervical intraepithelial neoplasia-1 (CIN1, 32.2%) to CIN2 (11.5%) and CIN3/CC (6.9%). The logistic regression model showed negative associations of anti-E5 antibodies with CIN1 (OR = 0.38), CIN2 (OR = 0.42), and CIN3/CC (OR = 0.32) groups, being statistically significant. Contrast analysis showed an inverse relationship between anti-E5 antibodies with HPV DNA and the CIN1 (OR = 0.35), CIN2 (OR = 0.39), and CIN3/CC (OR = 0.31) groups. Conclusions: These results suggest that anti-E5 antibodies could be associated with clearance of infection in women without lesions and with CIN1 lesions since an inverse relationship was observed between the presence of HPV DNA and anti-E5 antibodies. In contrast, with progression from CIN2/CIN3 to CC, the relationship was reversed, as the anti-E5 antibodies disappeared, and the frequency of the viral genome increased.

## 1. Introduction

Human papillomavirus (HPV) infections are a necessary cause but not sufficient to develop cervical cancer (CC) [1,2]. Women with persistent high-risk (HR) HPV infections, especially HPV16, are at increased risk of developing CC [2,3]. It is well documented that the prevalence of HPV increases with the severity of the disease, being between 46% in low-grade lesions (cervical intraepithelial neoplasia grade 1, CIN1) and up to 85% in CIN3. In invasive CC, the prevalence is over 99%, with HR-HPV types 16 and 18 being the more prevalent [4,5].

The HPV genome has the reading frame for the early proteins E1, E2, E4, E5, E6, and E7; late regions for L1 and L2; and the long control region that contains the origin of viral replication necessary for the regulation of viral gene expression. The E1 and E2 transcripts regulate the replication and transcription of viral DNA, and E4 helps in genome amplification and virus liberation [6]. E6 and E7 are the primary oncoproteins that are involved in the deregulation of the cell cycle through the degradation of p53 and the sequestration of pRB, respectively, while the E5 oncoprotein is involved in the early stage of the transformation process by regulating the epidermal growth factor receptor signal transduction cascade [7,8].

According to different studies, the HPV DNA identified in host cells is in episomal or integrated forms. Integration into the host cell genome suggests a critical step for cell transformation and cancer development [9,10]. Earlier studies have shown that viral genome integration alters the *E2* region and disrupts the *E5* gene transcription. This event suggests that E5 protein expression is lost in cancer progression and that this protein is involved mainly in the early steps of carcinogenesis [9,11]. However, it was demonstrated that HPV-positive tumors contain a mixture of episomal and integrated HPV DNA [11,12], which suggests that E5 could also play a role in the maintenance of the transformation stage during cancer development by driving cellular proliferation and apoptosis abrogation [13,14].

During HPV infection, a protective humoral immune response against L1 and L2 proteins is generated. Additionally, an antibody response is generated against E proteins, although these antibodies are not protective. Recently, it has been suggested that the antibody response against HPV proteins could be biomarkers for HPV-associated premalignant lesions and CC [15]. In this sense, antibodies against E6 and E7 proteins are the most frequently associated with CIN3, and CC [16,17]. Furthermore, the E4 protein (related to viral replication) is abundant in low-grade lesions, and the presence of anti-E4 antibodies has been associated with CIN1/2 lesions as well [18,19], suggesting that the E4 protein and/or the anti-E4 antibodies could be used as markers of low-grade lesions [15,20]. 

In the case of E5 protein, some research groups have looked for *E5* mRNA transcripts in premalignant and CC stages and found a high frequency of these transcripts (60% to 75%) in low squamous intraepithelial lesions (LSILs) and, with a heterogeneous frequency (30% to 70%), in high SILs (HSILs) [21,22], an event that has been associated with the presence of a mixture of episomal and integrated HPV DNA [23,24]. Furthermore, using an E5 peptide from HPV16, another research group demonstrated E5 expression in CC cell lines and, through organotypic raft cultures, showed high expression of E5 in LSIL-like 3D cultures and expression at lower levels in HSIL-like 3D cultures [25]. 

The difficulty in generating anti-E5 antibodies in the laboratory has hampered the identification of this viral protein in the natural history of CC and the study of the humoral immune response against it. To demonstrate the presence of E5 protein on cervical lesions, Chang’s group generated antibodies against HPV16 E5 protein and, using cervical biopsies, showed that this antigen was present with high frequency in LSILs (80%) and to a lesser extent in squamous cell carcinoma (SCC) (60%) [26]. More recently, serum antibodies against HR-HPV E proteins were characterized using protein microarrays, and anti-E5 antibodies were identified in the serum of patients with cervical lesions, but no associations were found with any specific stage of the disease [16,27].

In addition to CC, HR-HPV infections are associated with other types of cancer, such as anal, vulvar, oral, and head and neck cancer [28,29]. Concerning the E5 oncogene, two studies analyzed the presence of mRNA transcripts in oropharyngeal cancer (OPC) and demonstrated that *E5* transcripts were present in 75% of OPC samples and that the presence of this oncogene was correlated with better survival and absence of tumor recurrence in patients [30,31]. These observations suggest the critical role of E5 expression as a marker of lesion regression. By using anti-E5 antibodies as surrogate markers of the presence of the E5 oncoprotein, this study aimed to characterize the serum antibody response against HPV16 E5 in the natural history of CC, using the E5 protein, produced in vitro, in a slot blot system. Our results showed a high prevalence of anti-E5 antibodies in the NL (49.4%) and CIN1 (32.2%) groups. The anti-E5 antibodies also showed negative associations with the degree of cervical lesions and negative interactions with the presence of HPV DNA. These results suggest that anti-E5 antibodies could be associated with an active infection since an inverse relationship between the presence of HPV DNA and the progression of cervical lesions was observed when anti-E5 antibodies disappeared.

## 2. Materials and Methods

### 2.1. Study Population 

Women referred to the dysplasia clinic “Dr. Mauro Belauzaran Tapia” General Hospital in Cuautla, Morelos, Mexico, were invited to participate in the study. Informed consent was obtained from 335 women (mean age 40 years) diagnosed with abnormal histopathological study and 149 women who were finally diagnosed without cervical lesions and accepted to participate in the no lesion (NL) group (mean age 43.6 years). The women who agreed to participate signed an informed consent form and answered a questionnaire on variables related to sociodemographic and sexual behavior characteristics (age of first sexual intercourse, number of sexual partners, among others) and donated a blood sample to detect HPV16 E5 antibodies.

The presence of cervical lesions in the women who participated in the study was confirmed by histopathology and diagnosed as CIN1 (*n* = 215), CIN2 (*n* = 72), CIN3 (*n* = 34), and CC (*n* = 15) (Figure 1). Those women whose cervical lesions were confirmed by histopathology underwent treatment according to their medical doctors. The Ethical Committee from the National Institute of Public Health, Mexico, revised and approved this study (INSP-CI: 643, 23 January 2009).

The validation of the assay was carried out with samples of a sera bank from adolescents aged 9 to 13 years who had not initiated sexual relations and with a negative or low exposure to HPV. Dr. Eduardo C. Lazcano Ponce donated the negative control samples from a previous project that evaluated the immunogenicity of the HPV vaccine in Mexico (INSP-CI: 883, 13 October 2010). The Ethical Committee of the National Institute of Public Health revised and approved this study.

### 2.2. Production of HPV16 E5 Antigen and Analysis by Western Blot

The HPV16 E5 antigen with a histidine tag (His) was produced under in vitro conditions using the Rapid Translation System (RTS, Roche Molecular Biochemicals, Mannheim, Germany) following the manufacturer’s instructions. Briefly, the in vitro reaction was performed at 22 °C for 16 h with continuous stirring, and the recombinant E5 protein was purified by affinity chromatography on Ni-NTA (nickel-nitrilotriacetic acid) resin (Qiagen, Germantown, MD, USA) as described previously [20]. To verify the presence and identity of the HPV16 E5 protein, an anti-His monoclonal antibody generated in our laboratory was used to recognize the His-tag present in the recombinant E5 protein and analyzed by Western blot [32]. The antigen–antibody reaction was detected using the Western Lightning Chemiluminescence Reagent Plus (Perkin Elmer, Waltham, MA, USA), and the blots analyzed with the Odyssey Fc^®^ system (LI-COR Biosciences, Lincoln, NE, USA). The protein concentration of the purified E5 antigen was calculated from the immunoblots using a standard curve of a protein of known concentration and analyzed by densitometry using Image Studio Lite 4.0.21 software (Appendix A). The Green Fluorescent Protein (GFP) was produced under the same in vitro conditions as the E5 protein and used as a negative control of the system.

### 2.3. HPV Typing in Biological Samples

The pharmaceutical laboratory service LSG Clinicos (Mexicali, BC, Mexico) performed HPV genotyping within the cervical samples. PCR followed by pyrosequencing was performed to identify 17 genotypes (6, 11, 16, 18, 31, 33, 35, 39, 45, 51, 52, 53, 54, 56, 58, 59, 68). 

### 2.4. Slot Blot for Anti-E5 Antibody Detection

As previously reported, the slot blot assay for detecting anti-E5 antibodies was performed [20]. Briefly, E5 and GFP purified proteins (10 ng each) were placed onto a PROTRAN^®^ nitrocellulose membrane (Wathman Cytiva, Malborough, MA, USA) using the Hybrid-Slot™ Manifold (Biometra, Seattle, WA, USA). Sera samples (dilution 1:2500) were tested with strips containing specific E5 antigen, GFP antigen (negative control), and slot without antigen (background control). The slot blot system was enhanced using the biotin–streptavidin–HRP (horseradish peroxidase) (DAKO, Santa Clara, CA, USA) method, developed by chemiluminescence, and visualized in the Odyssey Fc^®^ system (LI-COR Biosciences, Lincoln, NE, USA). To record the optical density of the sera analyzed in the slot blot assays, Image Studio Lite 4.0.21 software was used (Appendix A).

Negative and positive control sera (previously analyzed for anti-E5 antibodies by ELISA) were introduced in the slot blot system for every 20 sera analyzed for system reproducibility. The cut-off point for total anti-E5 immunoglobulins was calculated from a titration antibody curve using the four-parameter equation described previously [20] and was expressed as Arbitrary Units (AU/mL). Briefly, serial dilutions of a pool of 10 anti-E5 positive sera were used to construct a titration curve, and the anti-E5-specific signal on the slot blot (pixels/area) was calculated (Appendix A). Sera from female adolescents (9 to 13 years) (*n* = 81) was used as a negative control to calculate the cut-off value. The cut-off point was defined as the geometric mean + 2SD, and this value was set at 5 AU/mL.

### 2.5. Statistical Analysis

A description of sociodemographic and sexual behavior characteristics of the NL group versus the CIN1, CIN2, and CIN3/CC groups was performed. Antibody levels against the HPV16 E5 antigen were measured in the serum of women with cervical lesions; the means were compared with the *t*-test, identifying the difference in proportions with chi-square. Logistic regression models were used to estimate odds ratios (ORs) with a 95% CI (Confidence Interval) for the association of anti-E5 serum antibodies (categorical variable) with risk factors, with premalignant lesions, and with CC, and adjusted by age. Furthermore, to determine the association of anti-E5 antibodies in the development of CC, a logistic regression model was used along with contrasts and linear hypothesis testing after estimation to evaluate the interaction of anti-E5 antibodies with the presence of HPV DNA and the degree of the cervical lesion. 

All the tests were two-sided, and the significance level was 5%. The statistical analysis was carried out using Stata 15 statistical software (Stata Corp, College Station, TX, USA).

## 3. Results

### 3.1. Characteristics of the Study Population

A comparison of the demographic and behavioral sexual characteristics of patients with different cervical lesions is summarized in Table 1. Overall, the mean age of the female population was 40 years old (range 18–64 years old). 

Still, when the variable was stratified, statistical differences were observed between the no lesion (NL) group and the CIN1, CIN2, or CIN3/CC groups (*p* = 0.002). These differences could be because the study population included a large group of women aged 30 to 49 years (64%; 310/485), which could influence these differences. No statistical differences were observed between the groups studied for the other demographic behavior characteristics analyzed (educational level, marital status, and smoking). For sexual behavior, we observed statistical differences between the groups with different numbers of pregnancies (*p* = 0.013), durations of sexual life (*p* = 0.001), numbers of sexual partners (*p* = 0.000), and types of infection (*p* = 0.002).

Therefore, and because it is well documented that age and sexual activity are risk factors for the acquisition of HPV infection, the statistical analysis was adjusted by these confounding variables [33,34].

### 3.2. High Prevalence and Levels of Anti-E5 Antibodies in Women with No Lesions and CIN1 Lesions

To characterize the anti-E5 antibodies in the serum of the female population, we used a standardized slot blot system for the E5 protein (produced under in vitro conditions). This system has demonstrated high sensitivity and specificity for other HPV antigens (E4, E7, and L1) [20]. Anti-E5 antibody levels (AU/mL) were calculated for each study group (as described in the Materials and Methods Section), and it was observed that the highest levels were present in the NL group, with mean levels higher than 5 AU/mL. However, these antibodies mean levels were lower (<3 AU/mL) in women with cervical lesions or CC, a difference that was statistically significant with the CIN1 group (*p* < 0.001) and to a lesser extent with the CIN2 group (*p* < 0.01) when compared to the NL group (Figure 2). 

The results shown in Table 2 are analyzed by column to show the distribution frequency of women positive for anti-E5 antibodies, according to the lesion grade category or HPV infection. The overall prevalence of anti-E5 antibodies in the study population was 17.9% (87/485). It was the NL group that had the highest prevalence (49.4%, 43/87). In contrast, the overall prevalence of anti-E5 antibodies in the studied population showed a downward trend in terms of disease progression: it was 32.2% (28/87) for CIN1, 11.5% (10/87) for CIN2, and only 6.9% (6/87) for the CIN3/CC group (Table 2).

### 3.3. Anti-E5 Antibodies Are Associated with the Degree of Cervical Lesion in the Female Population Studied

The associations of anti-E5 antibodies with the behavioral and sexual characteristics and the histopathological diagnoses of the women were analyzed by logistic regression (Table 2). From the overall risk factors studied, anti-E5 antibodies were associated only with the degree of the cervical lesion, although this was a negative association. The logistic regression model showed negative associations of anti-E5 antibodies with each one of the cervical lesion groups, CIN1 (OR = 0.38; 95% CI 0.22–0.67), CIN2 (OR = 0.42; 95% CI 0.19–0.91), and CIN3/CC (OR = 0.32; 95% CI 0.12–0.82), with all of them being statistically significant (*p* = 0.001, 0.029 and 0.018, respectively) (Table 2). A higher prevalence of anti-E5 antibodies was observed in the NL group, with a downward trend with the CIN1-3 and CC groups, which explains this result. 

On the other hand, when we analyzed the presence of anti-E5 antibodies vs. HPV infection, it was observed that most anti-E5-positive women were HPV-negative (79.3%, 69/87). The remaining 20.7% (18/87) of women were positive for anti-E5 and HPV, with 94.4% (17/18) positive for HR-HPV, and of these, 44.4% (8/18) were HPV16-positive. Logistic regression analysis did not show a statistically significant association between anti-E5 antibodies and HPV infection (Table 2).

In addition, association analysis was performed for anti-E5 antibodies and other risk factors, such as educational level, marital status, smoking, number of pregnancies, sexual partners, and duration of sexual life, and no statistically significant associations were observed.

### 3.4. Inverse Relationship of Anti-E5 Antibodies with HPV Positivity and the Degree of Cervical Lesions

To further characterize the negative association of anti-E5 antibodies observed with the different cervical lesions in the studied population, we evaluated the presence of HPV DNA and the degree of cervical lesion together with the presence of anti-E5 antibodies as another risk factor for the disease (Figure 3). When we evaluated the frequency of anti-E5 antibodies by type of lesion, the highest frequency was observed in the NL group (28.6%, 43/150), with a clear downward trend in the CIN1 (13.1%, 28/214), CIN2 (13.9%, 10/72), and CIN3/CC (12.2%, 6/49) groups. Analysis of the HPV DNA in the population by type of lesion showed an inverse behavior of the infection concerning anti-E5 antibodies since the lowest frequency was observed in the NL (14%, 21/150) and CIN1 (19.2%, 41/214) groups, with a clear upward trend for the CIN2 (27.8%, 20/72) and CIN3/CC (34.7%, 6/49) groups. Interestingly, an intersection between anti-E5 antibodies and HPV DNA was observed in CIN1 lesions, suggesting a possible interaction (Figure 3). 

Logistic regression with interaction contrast analysis was performed to further analyze the inverse relationship between anti-E5 antibodies and the presence of HPV DNA with the degree of cervical lesion (Table 3). The results showed a negative interaction between anti-E5 antibodies and the three groups of cervical lesions, CIN1, CIN2, and CIN3/CC (OR = 0.35, 0.39, and 0.31, respectively), all of them in HPV-negative women, a statistically significant event (*p* < 0.05). No interactions were observed between anti-E5 antibodies and the type of lesion in HPV-positive women (Table 3). This result demonstrates an inverse relationship between anti-E5 antibodies and HPV infection and the disease progression.

## 4. Discussion

The use of the humoral immune response against HPV proteins as a surrogate biomarker for the stage of infection and/or the stage of the cervical lesion is based on the fact that the sequential expression of early HPV proteins has been correlated in the cervix with serological data [15,35]. The advantage of using these serological biomarkers is that a blood sample is a less invasive technique; they can identify the type of infection (previous, present, or persistent) and be associated with the degree of the cervical lesion; and as the natural amplified response can detect low concentrations of HPV viral antigens. In this sense, several studies have demonstrated that antibodies against different viral antigens could be related to specific disease stages [15]. For instance, antibodies against the L1 protein are associated with previous exposure to HPV, the anti-E6 and anti-E7 antibodies are frequently associated with CIN3 and CC, and anti-E4 antibodies are present in women with CIN1/2 lesions [15,16,17,20]. Also, anti-E6 antibodies have been identified as predictive markers of disease in the case of oropharyngeal cancer as they can be detected as many as 10 years before the diagnosis of it, although this was not observed for CC [36,37]. In the case of antibodies against the E5 protein, there are only two reports where antibodies were tested against multiple HPV antigens, including E5, by using proteome microarrays with sera from women with cervical lesions; however, no association was identified between anti-E5 antibodies and HPV infection or any stage of the disease [16,27]. Other studies have been conducted to look for the relationship between the presence of E5 transcripts and the disease progression, but little evidence has been generated. Despite this, mRNA has been associated with low-grade lesions [21], while in biopsies of women with cervical lesions, the presence of E5 protein was associated with LSIL [26]. Given the lack of studies on this subject, it was of interest to determine if a new technique, such as the highly sensitive slot blot for anti-E5 antibodies, could detect them in the sera of women with premalignant cervical lesions.

In this study, our group characterized the antibody response against the HPV16 E5 oncoprotein in the natural history of cervical cancer. The results showed that there is a negative association and an inverse relationship between the presence of anti-E5 antibodies with the presence of HPV DNA and the degree of cervical lesion. The highest frequency of anti-E5 antibodies was observed in women without lesions, suggesting that these antibodies could be related to an HPV infection that has already been cleared, as the prevalence of HPV DNA in this group was the lowest. Also, women with CIN1/2 lesions showed a drop in anti-E5 antibody levels but increased HPV DNA. In this sense, our results agree with the study of Sahab and colleagues (2012), where they demonstrated that an LSIL-like cell line expressed higher levels of E5 protein than an HSIL-like cell line, showing an inverse association between the presence of the E5 protein and the degree of the lesion [25]. These results are also consistent with the integration of the HPV DNA with the cellular genome, inducing the loss of the *E2/E4/E5* region and disrupting the gene transcription [1]. This event suggests that E5 protein expression is lost at late stages of the disease (HSIL) and in CC [11], and a similar trend was observed with the anti-E5 antibodies in this study.

Similarly, our results showed an inverse relationship between the presence of anti-E5 antibodies and disease progression, as anti-E5 antibodies were present in the control group (NL) with a downward trend according to the degree of the lesion (from CIN1 to CIN2 and CIN3/CC). These results are also consistent with a previous study that found a downtrend of the E5 antigen as the lesions progress to cancer (LSIL, 80%; SCC, 60%) [26], which is a similar trend to that observed with anti-E5 antibodies and the degree of cervical lesions in our study. However, opposing results were recently reported by de Araújo and colleagues (2020), where the use of qPCR to detect HPV transcripts in tissue from cervical lesions showed that E5 mRNA levels were higher in HSILs and CC than in LSILs [38]. The differences between this study and ours could be because de Araújo and colleagues did not include a group without lesions, nor a large population of women with CIN1 and CIN2 lesions, which are the groups with the highest prevalence of anti-E5 antibodies and who may have high expression of E5 transcripts and protein. A more detailed study with a larger population of women without lesions and precancerous lesions would be of great interest to clarify these differences.

On the other hand, different studies have documented that during the integration of the HPV DNA into the cellular genome, the *E5* gene expression is lost. However, this integration is a rare event in CIN lesions. The most abundant mRNA transcripts in these lesions are E5 and E4, suggesting that the role of the E5 protein in the development of the disease occurs early during the infection, when the HPV DNA is episomal [11,12]. Our results agree with these findings, as the interaction of anti-E5 antibodies observed with the CIN1 lesions (a drop in anti-E5 antibody levels and an increase in HPV DNA) may be related to the integration process of the HPV genome during the progression of the disease. The HPV genome integration event could be associated with the progression of the lesion, and this could also result in the loss of *E5* gene expression, causing the absence of the viral antigen and probably the reduction in antibody levels, as was observed in our study.

The presence of HPV16 E5 mRNA and translated protein was identified in tumor-derived cell lines and cell lines harboring HPV16 episomal DNA, suggesting that early expression of the E5 protein during the viral cycle is required for the maintenance of episomal viral DNA [22,25]. These observations suggest that the E5 protein expressed early during the infection/transformation transition events could be a key marker of the early stage of the disease. Therefore, early generation of serum anti-E5 antibodies during HPV infection (anti-E5 present in NL and CIN1 groups) could be a surrogate biomarker of the E5 protein expression and, at the same time, suggests the presence of an active infection (presence of episomal HPV DNA because E5 is expressed). However, this hypothesis needs to be investigated in more detail. 

Finally, the inverse relationship observed between anti-E5 antibodies with HPV DNA and the early stage of the disease suggests that the integration of viral DNA during progression to CC leads to the loss of *E5* gene expression and, with it, to the reduction in anti-E5 antibody levels over time, allowing the progression to a high-grade lesion. Our results suggest that anti-E5 antibodies could be useful in identifying CIN1 lesions since the presence of HPV DNA and anti-E5 antibodies were observed at this early stage of the disease. 

In this study, we were able to detect a reduced number of HPV types with the genotyping system used (seven HR-HPV and two LR-HPV types). It is possible that some samples could be positive for other HPV types that were not detected in the test used or that the DNA detection test was not sensitive enough for the type of samples that were analyzed. However, the trend of prevalence of HPV16 by type of lesion observed in this study was similar to what was reported for Mexican women (NL 6%/3%; LSIL 7.5%/14.8%; HSIL 12.5%/15%; CC 26.5%/45%, respectively) [39]. Still, despite the low sensitivity of the HPV test, the results are reliable for the presence of anti-E5 antibodies as this is an amplified response to the presence of the E5 protein, and the slot blot was sensitive enough to detect it. Yet, it is clear that the sensitivity of the HPV detection system used in our study was low, and this is a limitation of our study.

This is the first study that described the antibody response against the E5 protein in women with cervical lesions, and this has been challenging because of the difficulty of generating the E5 antigen, but with the in vitro translation system and the high sensitivity of the new slot blot developed, it was possible to detect the antibody response against E5 protein. Moreover, the fact that the E5 protein is present at an early stage during the viral cycle may generate an amplified anti-E5 antibody response that could be useful as a marker to detect early cervical lesions associated with HPV, which could be missed during histopathological study as only a small part of the tissue is analyzed. This has been demonstrated for anti-E6 antibodies in oropharyngeal cancer [37]. However, further studies in a larger population of women with cervical lesions and follow-up for several years are required to better characterize the anti-E5 antibody response at this early stage of the disease.

## 5. Conclusions

CC and lesions related to it are preventable by HPV vaccination. However, the low coverage worldwide makes it hard to stop the appearance of new cases every year. For this reason, early detection of preneoplastic lesions associated with HPV infection is highly required. In this study, the results suggest that the presence of anti-E5 antibodies could be associated with the clearance of the infection in those with no lesions and CIN1 lesions since an inverse relationship between the presence of HPV DNA and the anti-E5 antibodies was observed in these groups. However, in the progression of cervical lesions CIN2/CIN3 to CC, the relationship was reversed as the anti-E5 antibodies disappeared with the progression of the disease, and the frequency of HPV DNA increased. More and larger studies are necessary to determine the usefulness of these anti-E5 antibodies as serological markers for the early detection of CC.

## Figures and Tables

**Figure 1 biomedicines-12-02699-f001:**
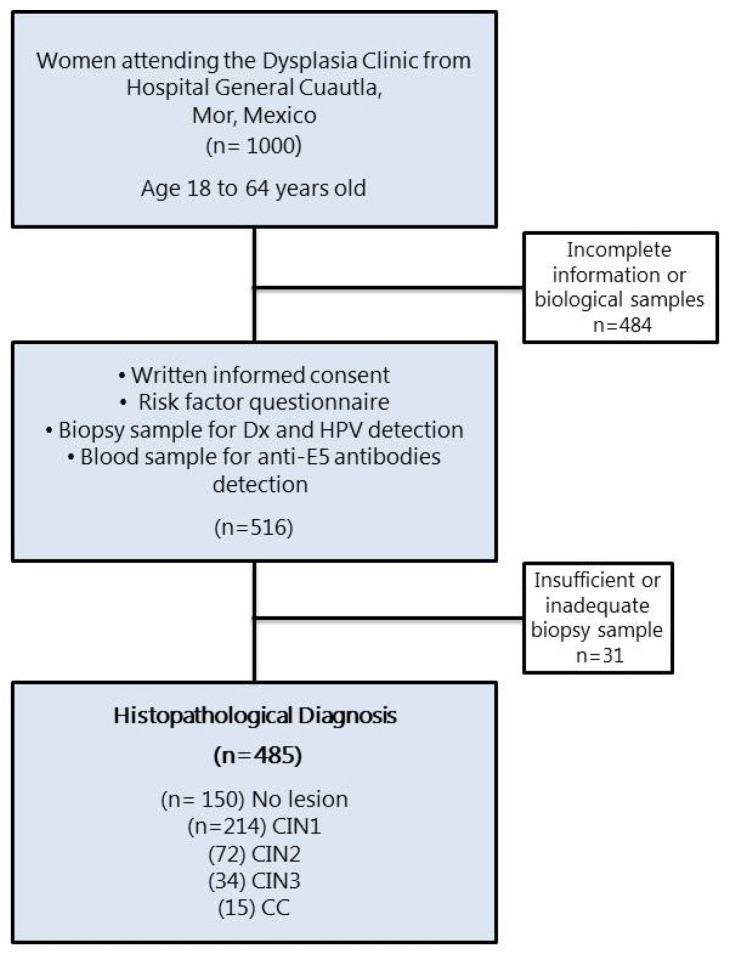
Flow diagram of the female population that participated in the study. CIN, cervical intraepithelial neoplasia grade 1, 2, or 3; CC, cervical cancer.

**Figure 2 biomedicines-12-02699-f002:**
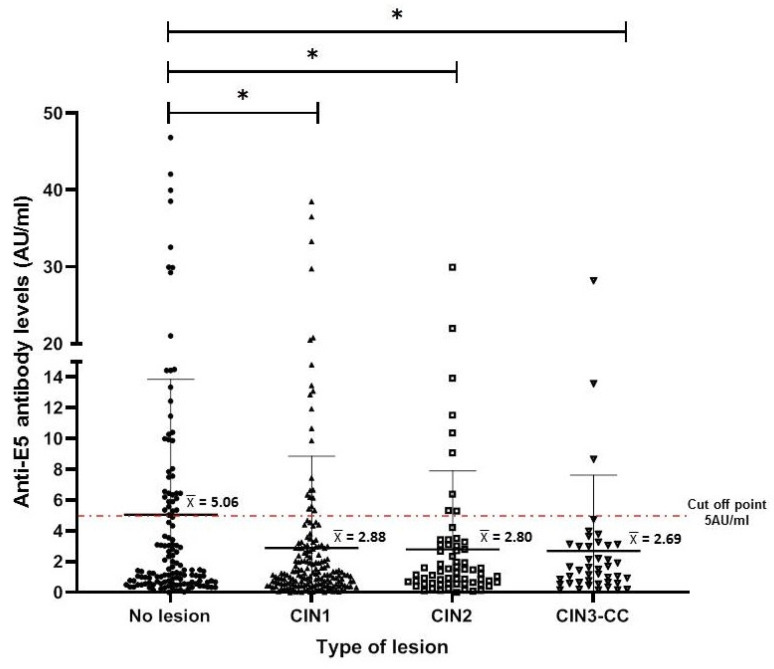
Anti-E5 antibody levels in women with different cervical lesions and CC. The median and interquartile range for each of the different cervical lesions are shown in the dispersion graph. The dashed line represents the cut-off point (5 AU/mL) as calculated for antibodies against E5 and described in Materials and Methods Section. The data were analyzed by the difference of medians using Kruskal–Wallis and Dunn testing, and statistical significances are shown as * *p* < 0.05.

**Figure 3 biomedicines-12-02699-f003:**
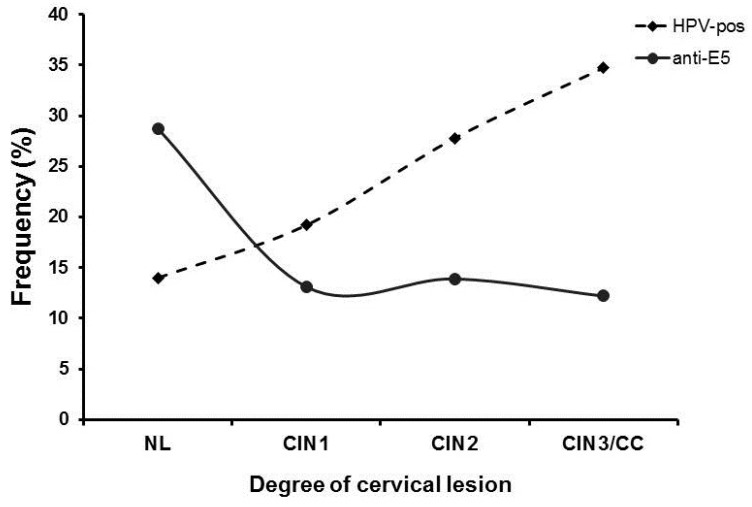
Frequency of anti-E5 antibodies and HPV DNA in women with different cervical lesions and CC. The frequency of women that were positive for anti-E5 antibodies and/or HPV DNA was plotted in a graph by the degree of the cervical lesion and analyzed for possible interactions. Anti-E5 antibodies (▪); HPV DNA (●).

**Table 1 biomedicines-12-02699-t001:** Demographic and sexual behavior characteristics of the study population.

Characteristic		No Lesion		CIN1			CIN2			CIN3/CC		
		*n* = 150		*n* = 214		*p* ^7^	*n* = 72		*p* ^8^	*n* = 49		*p* ^9^
		*n*	%	*n*	%		*n*	%		*n*	%	
**Demographics**												
Age	≤30 years	17	11.3	54	25.2	**<0.001**	15	20.8	0.123	7	14.3	0.398
	31 to 40 years	45	30.0	79	36.9		19	26.4		20	40.8	
	41 to 50 years	45	30.0	59	27.6		25	34.7		10	20.4	
	≥51 years	43	28.7	22	10.3		13	18.1		12	24.5	
Educational level	No education	32	21.3	43	20.1	**0.004**	20	27.8	0.569	12	24.5	0.852
	Basic	73	48.7	72	33.6		32	44.4		24	49.0	
	Medium–high	45	30.0	99	46.3		20	27.8		13	26.5	
Marital status	No steady partner ^1^	27	18.0	47	22.0	0.355	14	19.4	0.795	17	34.7	**0.014**
	Steady partner ^2^	123	82.0	167	78.0		58	80.6		32	65.3	
Smokes	No	119	79.3	181	84.6	0.196	63	87.5	0.138	36	73.5	0.390
	Yes	31	20.7	33	15.4		9	12.5		13	26.5	
**Sexual Behavior**												
Number of pregnancies	0–2	20	13.3	58	27.1	**0.002**	14	19.4	0.455	7	14.3	0.975
	3–6	76	50.7	106	49.5		32	44.4		24	49.0	
	≥7	54	36.0	50	23.4		26	36.1		18	36.7	
Duration of sexual life (in years) ^3^	≤15 years	33	22.0	81	37.9	**<0.001**	18	25.0	0.704	12	24.5	0.570
	16 to 34 years	83	55.3	115	53.7		41	56.9		23	46.9	
	≥35 years	34	22.7	18	8.4		13	18.1		14	28.6	
Number of sexual partners	0–1	88	58.7	126	58.9	0.968	39	54.2	0.526	17	34.7	**0.004**
	≥2	62	41.3	88	41.1		33	45.8		32	65.3	
Type of infection	Negative	129	86.0	173	80.8	0.522	52	72.2	0.090	32	65.3	**<0.001**
	Low-risk ^4^	2	1.3	7	3.3		1	1.4		1	2.0	
	High-risk ^5^	19	12.7	34	15.9		19	26.4		16	32.7	
	HVP16-pos ^6^	9	6.0	16	7.5		9	12.5		13	26.5	

^1^ Single women, widows, or divorcees; ^2^ married or with a stable partner; ^3^ obtained by subtracting the age of onset of sexual life from the age at the time of the study; ^4^ HPV6 and 11; ^5^ HPV16, 31, 33, 35, 52, 53, 54; ^6^ only HPV16-positive; ^7^ *p*-values calculated using chi-square (χ²) to compare NL with CIN1; ^8^ *p*-values calculated using chi-square (χ²) to compare NL with CIN2; ^9^ *p*-values calculated using chi-square (χ²) to compare NL with CIN3/CC. Significant *p* values between groups are marked in bold.

**Table 2 biomedicines-12-02699-t002:** Association of serum anti-E5 antibodies with disease diagnosis and HPV infection in the female population studied.

Characteristic		Anti-E5						
	Total *n* = 485	Negative *n* = 398		Positive *n* = 87		OR ^5^	CI (95%)	*p*
	*n*	*n*	% ^4^	*n*	% ^4^			
**Biopsy**								
No lesion	150	107	26.9	43	49.4	Ref.		
CIN1	214	186	46.7	28	32.2	0.38	(0.22–0.67)	**0.001**
CIN2	72	62	15.6	10	11.5	0.42	(0.19–0.91)	**0.029**
CIN3/CC ^1^	49	43	10.8	6	6.9	0.32	(0.12–0.82)	**0.018**
**HPV infection**								
Negative	386	317	79.7	69	79.3	Ref.		
Low-risk ^2^	11	10	2.5	1	1.2	0.54	(0.06–4.38)	0.57
High-risk ^3^	88	71	17.8	17	19.5	1.06	(0.58–0.94)	0.83
HPV16-pos	47	39	9.8	8	9.2	0.94	(0.41–2.14)	0.89

^1^ Women with CIN3 or cervical cancer; ^2^ HPV6 and 11; ^3^ HPV16, 31, 33, 35, 52, 53, 54; ^4^ percentages were calculated by column. ^5^ Logistic regression model adjusted for age; Ref. category used as a reference for the analysis. Significant *p* values between categories are marked in bold.

**Table 3 biomedicines-12-02699-t003:** Association of anti-E5 antibodies and HPV DNA according to the degree of cervical lesions and CC in the female population studied.

Type of Lesion	Total		Anti-E5				OR ^1^	(95% CI)	*p*
			Negative		Positive				
	*n* = 485	%	*n* = 398	%	*n* = 87	%			
**No lesion**									
HPV-Neg	121	27.0	94	23.6	37	42.5	Ref.		
HPV-Pos ^2^	19	3.9	13	3.3	6	6.9	1.06	(0.36–3.05)	0.91
**CIN1**									
HPV-Neg	180	37.1	158	39.7	22	25.9	0.35	(0.19–0.65)	**0.001**
HPV-Pos ^2^	34	7.0	28	7.0	6	6.9	0.50	(0.18–1.34)	0.173
**CIN2**									
HPV-Neg	53	10.9	46	11.6	7	8.1	0.39	(0.16–0.96)	**0.042**
HPV-Pos ^2^	19	3.9	16	4.0	3	3.5	0.29	(0.13–1.83)	0.293
**CIN3/CC**									
HPV-Neg	33	6.8	29	7.3	4	2.3	0.31	(0.10–0.97)	**0.045**
HPV-Pos ^2^	16	3.3	14	3.5	2	2.3	0.73	(0.14–3.72)	0.71

^1^ Logistic regression model adjusted for age; ^2^ positive for some of the high-risk HPV16, 31, 33, 35, 52, 53, 54. Ref. category used as a reference. Significant *p* values between categories are marked in bold.

## Data Availability

All the relevant data are accessible within the article and in the Appendix A.

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
