# Peer review of "Serum Antibodies Against the E5 Oncoprotein from Human Papillomavirus Type 16 Are Inversely Associated with the Infection and the Degree of Cervical Lesions"

_biomedicines, 2024, doi:10.3390/biomedicines12122699_

Round 1
Reviewer 1 Report (Previous Reviewer 1)
Comments and Suggestions for Authors
Review of the revised version of the manuscript by A. Salazar-Pina et al “Serological antibodies against the E5 oncoprotein from human papilloma virus type 16 are associated with early detection of uterine cervical lesions” Biomedicines 10.339
Manuscript by A Salarazar-Pina et al describes detection of anti-E5 antibodies in the sera of large cohort of women, healthy as well as with varying grades of cervical lesions including cervical cancer (CC). Authors start with description of the sociodemographic features of the cohort and positivity for DNA of high risk HPVs (hrHPVs). Correlatíons are found between socio demographic features and positivity for hrHPVs, such as duration of sexual life and number of sexual partners. Further authors describe methodology of detection of anti-E6 antibodies by slot blot. These parts of the manuscript have been improved.
Using slot blot, authors found healthy women to have the highest, and patients with CC, the lowest frequency of anti-E5 antibodies. No correlations were found between anti-E5 and majority of sociodemographic features of the studied cohort. On contrary, a strong inverse correlation of the presence of anti-E5 was found with the grade of lesions, which was further confirmed for HPV negative, but not in HPV positive women. This part was also improved.
Discussion has been improved as well, more issues included. However, conclusions remain basically unchanged, authors conclude that the antibodies are associated with early detection of cervical lesions (in the title) and acute HPV infection (in discussion).
Despite multiple improvements, I still have a number of serious concerns regarding this manuscript, now mostly to the discussion. Besides, there is no response of the authors to the 1st review, necessitating analysis of the changes made to meet the critics of the 1st review.
MAJOR from 1st review
1. FIRST REVIEW: Firstly, levels of anti-E5 antibodies are determined via quantification of the intensity of signals generated by patient sera on the NC membrane (slot blot). The NC strip contains E5, control protein GFP and buffer. Authors state that sera of women reacted with E5, did not react with buffer or GFP. There is no visual data to confirm, or just illustrate this. Supplement presents calibration curve of E7 stained with commercial anti-E7 mouse Ab (which does not react with buffer or GFP) – obviously to show how calibration curve should look like, but the curve is not for E5. No such curve is given for E5 anywhere. Furthermore, when they describe how E5 was loaded on NC, the write “ten microliters of E5 protein solution” which is not a way one quantifies the protein. Next, also in supplement, they show how a selection of E5 positive human sera reacts with E5, but they do not show how the same sera react to GFP or control – ie. do not show whole strips.
- SECOND REVIEW: Revision completed. Data provided in supplementary figures. However, suppl figures have not legends. Legends need to be provided.
2. FIRST REVIEW: Signal in slot blot is quantified against a calibration curve made by serial dilutions of 10 anti-E5 positive sera pools. It is not clear how many pools are used. It is not clear wherefrom these sera were taken – out of this study, or other studies. It is not clear how (against what controls) it was concluded that these sera have specific (NB!) anti-E5 reactivity? According to the data of these study, anti-E5 antibodies appear at the highest level in HPV negative women, they can possibly indicate HPV exposure, but cannot be associated to HPV infection. Hence, specific efforts have to be done to prove that they are specific, at least the sera which are used as positive controls. This should include blocking serum reactivity by E5 protein prior to Slot blot (to exhaust sera), and using serial dilutions of E5 on NC, to demonstrate loss of the staining with decreasing protein amount, with GFP diluted similarly as negative control, both preparations should contain known amount of proteins (E5 and GFP).
- SECOND REVIEW: Completed. Authors provide reactivity of positive and negative sera in Suppl Fig S2. However, they do not specify which sera are used – negative control – single sample, which one (out of the panel)? More important for positive, since authors were using not single sera, but a pool of 10 positive sera. They have to specify exactly what is shown on Suppl Fig. S2 – single positive serum sample, or a pool of 10 sera used as positive control throughout all assays. In Suppl Fig. S3 panel B, they need to specify that they show results of slot blot for the dilutions of the pool of anti-E5 antibody positive sera used as positive control.
3. FIRST REVIEW: Authors use GFP with His tag as a single negative control protein. GFP is logical to use, since it is prepared same way as E5. More control proteins have to be used, including ones used to identify sera with background unspecific antibody reactivities and polyreactivities (often, autoimmune) (see, for example, https://www.sciencedirect.com/science/article/pii/S152166160600115X ).
Additional consideration here relies to the nature of recombinant E5. Protein is produced in vitro and has His tag. That implies that reactivity is tested also against His-tag part of the protein. Polyhistidine sequences are present within a small number of human proteins and may direct expression to the nucleus and nuclear speckles compartments of the cell. Serum reactivity may result from crossrecognition of endogenous nuclear proteins (see for example https://pubmed.ncbi.nlm.nih.gov/24835186/ ). Recombinant GFP produced same way is used as a control, implying accounting for eventual reactivity against His tag in the sera. However, such His tag recognition depends also on the amino acid context in which His tag is placed. Additional control in Slot blot is needed of a synthetic peptide representing His tag with extra 2-3 amino acid on the terminus (termini) reproducing the site of His tag attachment in E5 (which do not represent a B-cell epitope by themselves).
SECOND REVIEW: Partially completed, authors show absence of reactivity of patient sera with His-tagged GFP, acceptable as control.
4. FIRST REVIEW: Cut-off value in reactivities is calculated against negative controls – female adolescents of 9-13 years. Wherefrom have these sera/these individuals came? They are not included into the study. What about ethics? What is meant by “naïve” (line 151)? The statement needs definition of the term and methodologically of the status. Where is the proof that these adolescents are/were not HPV infected? Infection is known to also occur by birth, via family contacts. Instead, authors just refer to “negative and positive control sera (previously characterized)” (line 145) without any explanations, or reference to earlier publications.
SECOND REVIEW: Information about the cohort of female adolescents included, their degree of HPV exposition/infection is addressed. OK
5. FIRST REVIEW: In Table 1, authors characterize the cohort by demographic features and behavioral characteristics. Table 1 shows no differences between the groups No lesion, CIN1, CIN2, CINS3-CC with respect to any factors, either demo, or social behavior. Further, no associations are described between these factors and positivity for HPV. However, authors introduce adjustment to age and sexual activity as known “risk factors for the acquisition if HPV infection”/”confounding variables” (lines 180-182), without any reference to either own, or published data.
SECOND REVIEW: Mostly addressed in revised Table 1 with references to earlier published data (lines 193-207). OK
6. FIRST REVIEW: Very strange figures on positivity for HR HPVs among women with CIN2 (26%), and even more so, CIN3 and CC (32,7%), for HPV16 26.5% Most of the studies published by now indicate that >90% of cervical cancer (and also CIN3) cases are associated with infection with one or more HR HPVs (see, for reference, https://www.cancer.gov/about-cancer/causes-prevention/risk/infectious-agents/hpv-and-cancer ). Figures presented by Salazar-Pina et al are very low, and raise questions on sensitivity of the tests, and with this, on the correct interpretation of occurrence of anti-E5 antibodies in relation to HR HPV positivity status.
SECOND REVIEW: Not addressed at all, no mentioning in the discussion.
7. FIRST REVIEW: Data in Table 2 is beyond understanding. I have started recalculating when I saw line “HPV infection”, “Negative”, n=386, negative 317, % 317/386=82,1% Authors write “79,7%”. OK but then positive are 69/386=17,9%, but authors write again, “79,3%”. After that, I started to look from the beginning: Line 1, Biopsy, patients without lesions, n=150. 107 are negative for anti-E5, % negative 107/150=71,3%. Authors write 26,9%, further, positive for antiE5 – n=43. 43/150=28,6%. Authors write 49.4%. Line 2 – total 214. Negative 186, in my understanding % is 186/214=86,9%. Authors write 46,7%. And these discrepancies follow throughout the table. If this is not the way they have counted %, they should have explained how this was done.
These insufficiencies and discrepancies in presentation of the data do not allow to interpret and meaningfully discuss the results. However, one can still make a general note.
SECOND REVIEW: Not addressed, figures and % in Table 2 remained unchanged without any explanations.
8. FIRST REVIEW: Interpretation of the data, starting from background is wrong, since (i) many cases of CIN2, CIN3, CC are associated with episomal HPVs, without integration, without loss of E5-encoding part of viral genome, and (ii) E5 has been repeatedly shown to be expressed also in CIN3 and CC. This is not discussed in the introduction. In view of the above findings, absence of antibodies reacting to E5 in CIN3 and CC can hardly be explained by the absence of circulating antigen available to the immune system of the patient. Last but not least, there is no data indicating that anti-E5 are correlated to acute HPV infection, as authors claim in the abstract.
SECOND REVIEW: Partially revised, but the conclusion is the same. Presence of anti-E5 Ab associates with acute infection – sound very strange as they are mostly observed in healthy not HPV infected individuals.
MINOR
9. FIRST REVIEW: To the title and throughout the text, authors use the term “serological antibodies”. What is meant? Antibody is a defined term, are antibodies found in sera any different from antibodies found elsewhere (or by means other than serology)?
SECOND REVIEW: Title changed, OK.
10. FIRST REVIEW: What is meant by “Time of sexual activity” (Table 1)? Duration of sexual life up to the quest?
SECOND REVIEW: Not done. Should be changes to “duration of sexual life (in years)”. “Time” is time point – if this term is used, it needs to be reformulated as “Start if sexual life, time/years back”.
11. FIRST REVIEW: Number of pregnancies is presented as 0-1 birth, 3-6 birth etc. Nn of pregnancies is not equal to number of births, not counted are spontaneous and planned abortions.
SECOND REVIEW: Corrected to pregnancies, OK.
12. FIRST REVIEW: Data on the strength of anti-E5 antibody reactivity in sera of healthy women, women with CIN1, CIN2 and CIN3+CC is illustrated in Fig 2 – showing loss of anti-E5 Ab levels (AU/ml). This cannot be called “titers” as in Fig 2, since authors have not titered the sera. What authors measure is the content of (or levels of) anti-E5 Ab in arbitrary units.
SECOND REVIEW: Changes not done, Y axis is still showing “titers”.
13. FIRST REVIEW: Seven out of 34 references (20%) are self-citations.
SECOND REVIEW: No changes in the reference list. Still 20% of self-citations, while many citations are missing in the text, to highly relevant publications, see secondary review below.
SECOND REVIEW NEW MAJOR CRITICS
#Review 2-1 Page 2, line 97 – authors state that they intend to use anti-E5 as surrogate markers of presence of E5 oncoprotein – and then they see max levels of these antibodies among healthy HPV negative women – how this statement can be linked to the observed phenomenon and authors explanations/discussion on their observations?
#Review 2-2 Table 1 – As in the 1st review: Very strange figures on positivity for HR HPVs among women with CIN2 (26%), and even more so, CIN3 and CC (32,7%), for HPV16 26.5% Most of the studies published by now indicate that >90% of cervical cancer (and also CIN3) cases are associated with infection with one or more HR HPVs (see, for reference, https://www.cancer.gov/about-cancer/causes-prevention/risk/infectious-agents/hpv-and-cancer ). Figures presented by Salazar-Pina et al are very low, and raise questions on sensitivity of the tests, and with this, on the correct interpretation of occurrence of anti-E5 antibodies in relation to HR HPV positivity status.
#Review 2-3 Further, Table 1 contains p values for comparison by Chi-squire test. The latter is for pair wise comparison. In Table 1, authors have 4 pt groups, and socio demographic features – 3 to 4 groupings per each feature. It is absolutely unclear what is compared to what, to what comparison – of which pairs – this p belongs? Should be indicated which pairs are compared, and p value given to compared pairs when they are different.
#Review 2-4 Section 3.2 presents data on high prevalence of anti-E5 in healthy individuals compared to patients, and their decrease in progressing from CIN1 to CINIII/CC. Further in discussion authors attribute this to decrease of E5 expression in CINIII and nearly absence of E5 expression in CINIII/CC which causes (gradually) decrease of the levels of circulating antibodies. Explanation could be quite different. Patients with solid cancers are known to be immunocompromised, specifically in ability to mount antibody responses. This had been repeatedly shown lately, in SARS-CoV-2 vaccine studies (see for example JAMA Oncol, 2021, 7(8), 1141-1148; or BMJ 2022; 376:e0686632). Levels of anti-E5 can reflect the same phenomenon, not decrease in anti-E5 due to fluctuations in the level of E5 protein. Ideal would be to determine levels of antibodies to common vaccines, or pathogens, as measles, or SARS-CoV-2 anti-N and anti-S proteins using commercial test systems, and compare the fluctuations of their levels with that of anti-E5. If this is not feasible, at least mention that reason for loss of anti-E5 could be gradual loss of immune competence with progression to cancer.
#Review 2-5 The 1st review pointed at very strange figures on positivity for HR HPVs among women with CIN2 (26%), and even more so, CIN3 and CC (32,7%), for HPV16 26.5% Most of the studies published by now indicate that >90% of cervical cancer (and also CIN3) cases are associated with infection with one or more HR HPVs (see, for reference, https://www.cancer.gov/about-cancer/causes-prevention/risk/infectious-agents/hpv-and-cancer ). Figures presented by Salazar-Pina et al are very low, and raise questions on sensitivity of the tests, and with this, on the correct interpretation of occurrence of anti-E5 antibodies in relation to HR HPV positivity status. This was not addressed at all, no mentioning in the discussion. Even in low resource settings 20 years ago, the hrHPV DNA was detected in 73.3% and 88.4% of 86 women with high-grade SIL or invasive cancer and in 12.2% of 2680 and 18.1% of 243 women without evidence of cervical disease ( https://doi.org/10.1093/jnci/92.10.818). Positivity for hrHPV DNA in CINIII/CC is still only 34,7% (page 8, lines 265-266) which strongly contradicts published data, and may indicate insufficient sensitivity of assays detecting hrHPV DNA.
#Review 2-6 Page 8, lines 279-281, state negative interaction between anti-E5 with severity of cervical lesions, all of which in HPV negative women, but not between anti-E5 and lesions in HPV positive women. These actually indicates that healthy women who have anti-E5 are less likely to be infected (as they are healthy and uninfected) and to develop lesions, i.e. they could be protective. This option needs to be considered alongside with the ones favored by the authors, that anti-E5 reflect encounter with hrHPV which (may) end up with lesions.
#Review 2-7 Discussion (page 9, line 305) cites use of anti-E6 antibodies are predictive cancer marker. For some reason, authors mention just one publication – for oropharyngeal cancer, not for cervical cancer, although there are plenty of solid studies on the subject – for example, Moller M et al, Virolology, 1992; Stanley M, Am J Obstetrics Gynecology 2003; Pal A, Kunda R, Frontiers Microbiol 2019, v10.
#Review 2-8 Page 11, lines 372 and further – the study does not describe Ab response to E5 in women with cervical lesions – rather on contrary – loss of anti-E5 with development of cervical lesions. Amplified anti-E5 response could not be used to detect precancerous lesions, as it disappears with lesions progression. Statement of the authors regarding predictive value of anti-E5 Ab levels can surely be validated in further studies, if authors perform follow up of healthy women with anti-E5 and find that the ones with early anti-E5 Ab later develop lesions, but not in any other circumstances.
SECOND REVIEW MINOR CRITICS
The ones left unaddressed from Review 1 - see comments to minor critics of the 1st round review). Additional comments:
Page 2, lines 55-63 – mixture of statements concerning acute and chronic HPV infection. Specifically lines 61-63 – starting with HPV positive tumors, and ending with the role of E5 during early stages of cancer development.
Page 2, line 64 – “during HPV infection, a protective humoral immune response against L1 and L2 is generated”. It is accepted that HPV infection does not induce a protective immune response (see for example https://www.sciencedirect.com/science/article/abs/pii/B9780128144572000180 ). It is induced only in vaccination.
Page 2, line 67 – “uterine” – authors need to adhere to one and the same terminology – “cervical”.
Page 3, section 2.1 – lines 122 and 128 with reference to ethical permits need dates to when permits were given.
Page 7, Table 2 – Table legend needs to explain what is compared to what – p value reflects comparison of what - no lesion to CIN1, then CIN2, then CIN3/CC?
Author Response
See the attched file for the point by point response to the reviewer´s comments

Reviewer 2 Report (Previous Reviewer 2)
Comments and Suggestions for Authors
The authors have addressed all my concerns adequately and I think that the manuscript can be published in the current form.
Author Response
No furhter comments were required for this reviewer.
Round 2
Reviewer 1 Report (Previous Reviewer 1)
Comments and Suggestions for Authors
The authors have considerably improved the manuscript answering the bulk of critical issues. I still do not agree with their concept on the nature of anti-E5 antibodies and their role, but now authors list different options including their original explanation that anti-E5 are markers of early stage of neoplastic lesions. Since other options are also presented, the reader can choose which to accept.
This manuscript is a resubmission of an earlier submission. The following is a list of the peer review reports and author responses from that submission.
Round 1
Reviewer 1 Report
Comments and Suggestions for Authors
Review of manuscript by A. Salazar-Pina et al “Serological antibodies against the E5 oncoprotein from human papilloma virus type 16 are associated with early detection of uterine cervical lesions” Biomedicines 10.339
Manuscript dby A Salarazar-Pina et al describes detection of anti-E5 antibodies in the sera of large cohort of women, healthy as well as with varying grades of cervical lesions including cervical cancer (CC). Authors find healthy women to have the highest, and patients with CC, the lowest frequency of anti-E5 antibodies. They describe an inverse correlation of the presence of anti-E5 with the grade of lesions in HPV negative women, and no correlation of anti-E5 with severity in HPV DNA positive women (page 7, lines 244-248). From this, they somehow conclude that the antibodies are associated with the acute HPV infection (in the Abstract) and detection of uterine lesions (title).
I have a series of major concerns regarding these data and this manuscript.
MAJOR
1. Firstly, levels of anti-E5 antibodies are determined via quantification of the intensity of signals generated by patient sera on the NC membrane (slot blot). The NC strip contains E5, control protein GFP and buffer. Authors state that sera of women reacted with E5, did not react with buffer or GFP. There is no visual data to confirm, or just illustrate this. Supplement presents calibration curve of E7 stained with commercial anti-E7 mouse Ab (which does not react with buffer or GFP) – obviously to show how calibration curve should look like, but the curve is not for E5. No such curve is given for E5 anywhere. Furthermore, when they describe how E5 was loaded on NC, the write “ten microliters of E5 protein solution” which is not a way one quantifies the protein. Next, also in supplement, they show how a selection of E5 positive human sera reacts with E5, but they do not show how the same sera react to GFP or control – ie. do not show whole strips.
2. Signal in slot blot is quantified against a calibration curve made by serial dilutions of 10 anti-E5 positive sera pools. It is not clear how many pools are used. It is not clear wherefrom these sera were taken – out of this study, or other studies. It is not clear how (against what controls) it was concluded that these sera have specific (NB!) anti-E5 reactivity? According to the data of these study, anti-E5 antibodies appear at the highest level in HPV negative women, they can possibly indicate HPV exposure, but cannot be associated to HPV infection. Hence, specific efforts have to be done to prove that they are specific, at least the sera which are used as positive controls. This should include blocking serum reactivity by E5 protein prior to Slot blot (to exhaust sera), and using serial dilutions of E5 on NC, to demonstrate loss of the staining with decreasing protein amount, with GFP diluted similarly as negative control, both preparations should contain known amount of proteins (E5 and GFP).
3. Authors use GFP with His tag as a single negative control protein. GFP is logical to use, since it is prepared same way as E5. More control proteins have to be used, including ones used to identify sera with background unspecific antibody reactivities and polyreactivities (often, autoimmune) (see, for example, https://www.sciencedirect.com/science/article/pii/S152166160600115X ).
Additional consideration here relies to the nature of recombinant E5. Protein is produced in vitro and has His tag. That implies that reactivity is tested also against His-tag part of the protein. Polyhistidine sequences are present within a small number of human proteins and may direct expression to the nucleus and nuclear speckles compartments of the cell. Serum reactivity may result from crossrecognition of endogenous nuclear proteins (see for example https://pubmed.ncbi.nlm.nih.gov/24835186/ ). Recombinant GFP produced same way is used as a control, implying accounting for eventual reactivity against His tag in the sera. However, such His tag recognition depends also on the amino acid context in which His tag is placed. Additional control in Slot blot is needed of a synthetic peptide representing His tag with extra 2-3 amino acid on the terminus (termini) reproducing the site of His tag attachment in E5 (which do not represent a B-cell epitope by themselves).
4. Cut-off value in reactivities is calculated against negative controls – female adolescents of 9-13 years. Wherefrom have these sera/these individuals came? They are not included into the study. What about ethics? What is meant by “naïve” (line 151)? The statement needs definition of the term and methodologically of the status. Where is the proof that these adolescents are/were not HPV infected? Infection is known to also occur by birth, via family contacts. Instead, authors just refer to “negative and positive control sera (previously characterized)” (line 145) without any explanations, or reference to earlier publications.
5. In Table 1, authors characterize the cohort by demographic features and behavioral characteristics. Table 1 shows no differences between the groups No lesion, CIN1, CIN2, CINS3-CC with respect to any factors, either demo, or social behavior. Further, no associations are described between these factors and positivity for HPV. However, authors introduce adjustment to age and sexual activity as known “risk factors for the acquisition if HPV infection”/”confounding variables” (lines 180-182), without any reference to either own, or published data.
6. Very strange figures on positivity for HR HPVs among women with CIN2 (26%), and even more so, CIN3 and CC (32,7%), for HPV16 26.5% Most of the studies published by now indicate that >90% of cervical cancer (and also CIN3) cases are associated with infection with one or more HR HPVs (see, for reference, https://www.cancer.gov/about-cancer/causes-prevention/risk/infectious-agents/hpv-and-cancer ). Figures presented by Salazar-Pina et al are very low, and raise questions on sensitivity of the tests, and with this, on the correct interpretation of occurrence of anti-E5 antibodies in relation to HR HPV positivity status.
7. Data in Table 2 is beyond understanding. I have started recalculating when I saw line “HPV infection”, “Negative”, n=386, negative 317, % 317/386=82,1% Authors write “79,7%”. OK but then positive are 69/386=17,9%, but authors write again, “79,3%”. After that, I started to look from the beginning: Line 1, Biopsy, patients without lesions, n=150. 107 are negative for anti-E5, % negative 107/150=71,3%. Authors write 26,9%, further, positive for antiE5 – n=43. 43/150=28,6%. Authors write 49.4%. Line 2 – total 214. Negative 186, in my understanding % is 186/214=86,9%. Authors write 46,7%. And these discrepancies follow throughout the table. If this is not the way they have counted %, they should have explained how this was done.
These insufficiencies and discrepancies in presentation of the data do not allow to interpret and meaningfully discuss the results. However, one can still make a general note.
8. Interpretation of the data, starting from background is wrong, since (i) many cases of CIN2, CIN3, CC are associated with episomal HPVs, without integration, without loss of E5-encoding part of viral genome, and (ii) E5 has been repeatedly shown to be expressed also in CIN3 and CC. This is not discussed in the introduction. In view of the above findings, absence of antibodies reacting to E5 in CIN3 and CC can hardly be explained by the absence of circulating antigen available to the immune system of the patient. Last but not least, there is no data indicating that anti-E5 are correlated to acute HPV infection, as authors claim in the abstract.
MINOR
9. To the title and throughout the text, authors use the term “serological antibodies”. What is meant? Antibody is a defined term, are antibodies found in sera any different from antibodies found elsewhere (or by means other than serology)?
10. What is meant by “Time of sexual activity” (Table 1)? Duration of sexual life up to the quest?
11. Number of pregnancies is presented as 0-1 birth, 3-6 birth etc. Nn of pregnancies is not equal to number of births, not counted are spontaneous and planned abortions.
12. Data on the strength of anti-E5 antibody reactivity in sera of healthy women, women with CIN1, CIN2 and CIN3+CC is illustrated in Fig 2 – showing loss of anti-E5 Ab levels (AU/ml). This cannot be called “titers” as in Fig 2, since authors have not tittered the sera. What authors measure is the content of (or levels of) anti-E5 Ab in arbitrary units.
13. Seven our of 34 references (20%) are self-citations.
Comments on the Quality of English LanguageQuality of English language is OK.
Reviewer 2 Report
Comments and Suggestions for Authors
The manuscript presented for review is an interesting analysis of the possible role of anti-E5 antibodies (as substitution for actual determination of E5 protein expression) as a biomarker for early cervical lesions.
The manuscript is well written and contains the well designed and conducted research. My suggestion to the authors are not critical but would higly improve the presentation:
1. Please remove the word "uterine" from the title. Cervical is sufficient to adequately describe the studied lesions. Uterine only adds confusion.
2. Please reformat the tables not as a figure but rather as tables (the quality is diminished in the current for and numbers are uclear.
3. In the flow-chart there is a small mistake - when adding the numbers of patients with lesions we get 335 not 336 as stated. Please correct the error.
4. Minor English editing needed.
5. Please add a paragraph describing the E protein family and their role in oncogenesis
6. Please give more information on the results of similar studies available in literature.
7. Please add strenghts and limitation paragraph at the end of Discussion section.
8. The conclusions are mostly about HPV vaccination which is not discussed in text. Please discusse it in text and provide proposed methodology for linking HPV vaccination to levels of serum anti-E5 antibodies - clinical usage.
Comments on the Quality of English LanguageMinor english proofing needed.